# Opportunities and Challenges of Frontier Data Governance With Synthetic Data

## Abstract

Synthetic data, or data generated by machine learning models, is increasingly emerging as a solution to the data access problem. However, its use introduces significant governance and accountability challenges, and potentially debases existing governance paradigms, such as compute and data governance. In this paper, we identify 3 key governance and accountability challenges that synthetic data poses - it can enable the increased emergence of malicious actors, spontaneous biases and value drift. We thus craft 3 technical mechanisms to address these specific challenges, finding applications for synthetic data towards adversarial training, bias mitigation and value reinforcement. These could not only counteract the risks of synthetic data, but serve as critical levers for governance of the frontier in the future.

## 1 Introduction

The rapid advancement of AI has led to an impending data bottleneck, where frontier models require exponentially increasing volumes of high-quality data, with some suggesting that the size of training corpora may exceed the sum of all human-generated data by 2030 (Villalobos et al., 2024). Empirical model capability scaling laws dictate that this scarcity bounds the overall capabilities of the frontier, regardless of strides in algorithmic complexity or computational power (Ruan et al., 2024), thus positioning it as a key issue for model developers to address. In the long term, data-efficient architectures may emerge in response to this problem and come to define the frontier. However, in the short term, it appears that synthetic data, or data generated by machine learning models as opposed to by humans, is increasingly defining the frontier. Indeed, already, a large portion of the training data of leading models in synthetic (OpenAI et al., 2024).

This shift towards synthetic data is one of many evolutions in the way models are trained that may jeopardize the efficacy of current approaches to governing and ensuring trustworthiness on the frontier. Compute governance, as proposed by Sastry et. al., for example, regulates computational power as a proxy for model capabilities (Sastry et al., 2024), but the link between the two grows more tenuous in the face of compute-efficient architectures and distilled models (DeepSeek-AI et al., 2025). Similarly, data governance, as proposed by Hausenloy et. al., relies partially on governing the flow of data through "AI Data Supply Chain" (Hausenloy et al., 2024), which they note becomes degenerate with regards to synthetic data, where the data generators, processors and trainers are the same people.

However, we propose that the advent of synthetic data, instead of being a limit to governance efforts, offers unique opportunities as a regulatory lever in addition to its challenges. Indeed, these may serve to be critical vectors for model control in the future, as existing approaches grow less effective, as detailed above.

Specifically, we identify 3 key challenges that synthetic data may pose to governance and accountability initiatives, and craft technical mechanisms to not only counter them, but establish synthetic data as a robust lever for governance of the frontier.

## 2 RELATED WORK

This paper is in the style of similar proposals for governance paradigms, such as the aforementioned "compute governance" and "data governance" (Sastry et al., 2024; Hausenloy et al., 2024). Specifically, it builds upon this work by addressing a technical and temporal hole in the previous paradigms - how they will apply to synthetic data specifically, and the future more generally.

Our work lies within the subfield of Technical AI Governance (Reuel et al., 2024), as its primary application is translational: proposing technical mechanisms for policy applications, while simultaneously providing a roadmap for technical and policy work from a governance perspective.

### CHALLENGES

We outline 3 key challenges synthetic data poses to governance and accountability frameworks

### 2.1 SYNTHETIC DATA CAN BE USED TO GENERATE MISALIGNED DATA AT SCALE.

The same ability to produce vast, tailored examples that makes synthetic data attractive to model trainers makes synthetic data attractive to malicious actors. Instead of using traditional, transparent data pipelines (Steinhardt et al., 2017), adversaries can mass-produce skewed data to deliberately misinform models (Biggio et al., 2012; Jagielski et al., 2018). Without proper safeguards, such "data poisoning" or "value hijacking" can lead to harmful ideologies or unreliable predictions in critical sectors like healthcare, finance, or public policy (Bender et al., 2021; Carlini et al., 2019).

### 2.2 SYNTHETIC DATA CAN DETACH MODELS FROM REAL-WORLD CONTEXTS.

Rich synthetic environments, while useful for scalable training, risk insulating models from the dynamic value signals present in authentic human interactions. Without continual exposure to genuine linguistic subtleties, cultural norms, and ethical considerations, models may develop value systems that diverge from societal expectations (Shrivastava et al., 2017; Torralba & Efros, 2011). This insulation is further exacerbated by feedback loops where models are retrained on their own synthetic outputs, potentially entrenching misaligned values over time (Richter et al., 2020; Ganin et al., 2016).

### 2.3 SYNTHETIC DATA COULD LEAD TO SPONTANEOUS BIASES IN BLACK-BOX SYSTEMS.

When large models are repeatedly retrained on their own synthetic outputs, the inherent opacity of deep learning architectures can allow small biases to accumulate unpredictably (Mehrabi et al., 2021). Over time, these biases may distort model outputs and compromise fairness, yielding results that conflict with societal expectations (Caliskan et al., 2017; Blodgett et al., 2020; Bender et al., 2021).

## 3 OPPORTUNITIES AND MECHANISMS

We propose 3 mechanisms to counter the challenges outline above.

### 3.1 SYNTHETIC DATA FOR ADVERSARIAL TRAINING

Synthetic data for adversarial training offers a scalable approach to enhance the robustness and safety of large-scale models by systematically generating malicious or deceptive scenarios. This counteracts the challenge of synthetic data for misaligned data generation. These scenarios can be used to identify and correct weaknesses in the model that might be exploited in real-world attacks. By synthetically creating adversarial examples at scale, researchers and practitioners can refine model behavior post-training, ultimately contributing to more secure frontier AI systems.

**Implementation:** To incorporate synthetic adversarial data in a training pipeline, one first delineates the scope of potential attacks (e.g., specific perturbations, semantic manipulations, or deceptive prompts). A generative model such as a Variational Autoencoder or a diffusion-based generator can

then be trained on a seed corpus of real-world examples, introducing adversarial constraints during data generation. This yields a large corpus of synthetic adversarial samples. These samples are integrated into the fine-tuning phase of the model, where iterative testing and updating ensure that previously uncovered weaknesses are addressed. Over multiple training rounds, the model learns to withstand a diverse set of adversarial inputs.

**Existing work:** Past research on adversarial training has largely relied on perturbations of real data (Goodfellow et al., 2015; Madry et al., 2018), but recent work has shown promise in generating synthetic adversarial inputs using Generative Adversarial Networks (GANs) (Creswell et al., 2018) or large-scale language models (Brown et al., 2020). Other studies have leveraged simulation frameworks and domain-specific generative models (Kingma & Welling, 2014; Dhariwal & Nichol, 2021) to produce highly varied adversarial examples that mimic real-world conditions. These approaches indicate that synthetic data can be a powerful tool in building adversarially robust systems, freeing the model developer from reliance on exhaustively labeled, human-crafted attacks.

**Challenges and mitigation:** While synthetic adversarial data broadens the space of potential attacks, it may also introduce novel biases if the generation process is insufficiently diverse or guided by incomplete threat models. Maintaining alignment between synthetic data distributions and real-world attack vectors can be difficult, requiring continuous monitoring and updating of generative pipelines. Additionally, the iterative feedback loop—whereby models trained on synthetic adversarial data might in turn generate subsequent synthetic data—demands careful oversight to prevent the accumulation of unrealistic or unrepresentative scenarios. Despite these challenges, synthetic adversarial data remains a valuable strategy for improving model robustness and proactively defending against the evolving landscape of security threats.

## 3.2 SYNTHETIC DATA FOR BIAS MITIGATION

**Motivation:** Real-world training datasets often suffer from demographic imbalances, such as underrepresenting certain regions or populations. For instance, health records might primarily originate from areas with highly developed healthcare systems, skewing predictive models toward those populations (Obermeyer et al., 2019). This entrenches the risk of differential treatment and can perpetuate inequities in service and care, as frontier AI systems learn more effectively from the demographics on which they have better data.

**Implementation:** Synthetic data can help mitigate these disparities by programmatically generating representative samples from underrepresented groups. This counteracts the challenge of spontaneous biases in synthetic data. One approach involves using generative models trained on smaller, high-quality samples from the minority population, then augmenting them with carefully designed synthetic instances (Chawla et al., 2002). In healthcare contexts, for example, generative adversarial networks have been employed to produce synthetic electronic health records that capture complex, multi-label characteristics (Choi et al., 2017). Additionally, domain experts and local stakeholders should guide the synthetic data generation process to ensure cultural and contextual fidelity.

**Existing work:** A growing body of literature highlights the use of generative techniques to correct or compensate for dataset biases. For example, creating synthetic faces using state-of-the-art generative adversarial networks has been shown to improve classification accuracy for underrepresented groups (Karras et al., 2019), and similar data augmentation strategies have been applied to textual data to enhance model performance (Wei & Zou, 2019). Moreover, frameworks like FairGAN have been developed to generate fairness-aware synthetic data that directly address biases in the training set (Xu et al., 2019).

**Challenges and mitigation:** Although synthetic data offers a promising avenue for reducing bias, it can also inadvertently introduce new biases or distort real-world distributions. Over-reliance on artificially constructed examples may yield models that perform poorly under complex, real-world conditions. Continuous monitoring, along with rigorous validation against real data, is critical. Furthermore, transparent documentation of synthetic data generation—outlining assumptions, constraints, and potential sources of error—can help stakeholders trust and verify the final models' fairness and efficacy (Mehrabi et al., 2021).

## 3.3 Synthetic Data for Value Reinforcement

**Motivation:** Large-scale AI models are increasingly vulnerable to data poisoning and value hijacking, wherein adversarial actors inject harmful ideologies or manipulative content into open-source training corpora (Biggio et al., 2012; Steinhardt et al., 2017). Such attacks can distort a model's values, nudging its decisions or outputs toward harmful agendas. By contrast, synthetic data generation provides an opportunity to purposefully curate the values embedded in a training set. This counteracts the challenge of synthetic data leading to detached environments. Rather than indiscriminately scraping the web—where harmful, misleading, or biased content may dominate (Bender et al., 2021)—lab-curated synthetic corpora can emphasize collaborative, ethical, and socially constructive values.

**Implementation:** To implement value reinforcement via synthetic data, developers can design generative models or specialized data augmentation pipelines that focus on producing content aligned with a set of predefined principles. For instance, a language model might be guided to generate texts that uphold specific ethical frameworks or emphasize fairness and respect across different cultural perspectives (Ziegler et al., 2019). This process can include the following steps:

1. *Define Value Targets:* Collaborate with ethicists, domain experts, and stakeholders to outline desirable attributes and behaviors, translating them into clear guidelines for synthetic data generation (Amodei et al., 2016).

2. *Curated Seed Data:* Compile a smaller, high-quality corpus exemplifying the targeted values. This set serves as the seed for training or fine-tuning a generative model.

3. *Generative Pipeline:* Employ large language models, diffusion-based methods, or other generative frameworks to produce synthetic samples that faithfully reflect the curated seed's values. Mechanisms such as reinforcement learning or policy gradients can ensure alignment with these standards (Christiano et al., 2017).

4. *Validation and Iteration:* Validate generated content against established guidelines. Discard or correct any synthetic instances that deviate from the desired value set. Iteratively retrain or fine-tune the model as needed (Gehman et al., 2020).

**Incentives for AI Labs:** Beyond ethical considerations, AI developers have pragmatic reasons to invest in value-aligned synthetic data. Models trained on carefully curated content often demonstrate higher-quality outputs, more robust performance, and fewer public-relations liabilities. By proactively filtering out harmful or adversarial material, labs can mitigate reputational risks, reduce moderation overhead, and foster user trust. As a result, curation becomes more than a moral imperative—it is also a strategic advantage.

**Challenges and mitigation:** Achieving broad consensus on which values to promote can be contentious, particularly when cultural, political, or organizational perspectives diverge. Additionally, overly restrictive curation may limit the model's exposure to diverse viewpoints, potentially compromising its adaptability or realism. Regular review by multidisciplinary teams can help calibrate the balance between value alignment and open-world robustness. Finally, just as data poisoning can subvert open datasets, sophisticated attackers may attempt to introduce subtle biases into curated pipelines, necessitating continual monitoring, audits, and transparency in the curation process.

## Conclusion

Synthetic data offers a powerful yet double-edged solution for frontier AI. It can overcome data scarcity and enhance model robustness, but without proper oversight, it risks fostering misaligned values and entrenched biases. The future of synthetic data in AI governance depends on innovative oversight mechanisms and transparent, collaborative frameworks that ensure its benefits are realized without compromising ethical standards.

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
