# OpenReview forum: "Opportunities and Challenges of Frontier Data Governance With Synthetic Data"
_ICLR.cc/2025/Workshop/BuildingTrust — Submitted to BuildingTrust_

### Official Review · Reviewer_Ya85 · 2025-02-18
**The identified challenges are trivial**

**Rating:** 5
**Confidence:** 4

**Review:**

- The identified challenges are trivial and applicable to any training setting, not specific to synthetic data. Poisoning, lack of data diversity, and bias are well-known and well-studied issues.
- The authors argue that their advantage over prior work is technical solutions. However, the solutions are speculative, and there are no theoretical or empirical validation. The challenges introduced are too complicated to be addressed through speculations.

---

### Official Review · Reviewer_YBRs · 2025-02-28
**A valuable starting point for discussions on synthetic data governance but requires substantial revisions before publication**

**Rating:** 4
**Confidence:** 4

**Review:**

# Significance
This paper explores the opportunities and challenges of frontier data governance using synthetic data. It identifies three key governance and accountability challenges: the increased emergence of malicious actors, spontaneous biases and value drift. These challenges are highly relevant to ongoing discussions in both technical and regulatory domains.

# Overall Quality and Evaluation
The paper is well-written and addresses an important topic. However, its motivation and methodology lack clarity. Specifically:

- The methodology for selecting the three identified challenges is not well explained.
- While the paper claims to identify three opportunities, these are not clearly presented.
- The authors do not adequately connect the identified opportunities to existing governance mechanisms and frameworks. As a result, it remains unclear how these mechanisms can serve as "critical levers for governance," as the authors suggest.

# Suggestions for Improvement
The governance of synthetic data intersects with multiple domains, including data, model, and compute governance. Attempting to cover such a broad scope within a four-page paper leads to gaps in the discussion. To improve clarity and depth, I suggest the following:

- Focus on a specific sectoral use case, such as healthcare, where synthetic data governance is particularly critical. While Section 3.2 briefly discusses synthetic data in healthcare, it is the only concrete example in the paper. Expanding on this or choosing another well-defined use case would strengthen the analysis.
- Select and elaborate on specific governance characteristics (e.g., key actors such as data controllers, challenges like replicability, and governance mechanisms such as model security). A more structured approach would enhance the paper’s contribution.

---

### Official Review · Reviewer_gfTU · 2025-03-01
**Review of "Opportunities and Challenges of Frontier Data Governance With Synthetic Data"**

**Rating:** 6
**Confidence:** 3

**Review:**

The paper summarizes the challenges and opportunities associated with using synthetic training data for large machine learning models. It focuses on three main aspects: (a) Synthetic adversarial examples, which can both perturb and robustify a model.
(b) Unskewed synthetic data, which may help mitigate biases but can also introduce new and different biases. (c) Synthetic data that adheres to agreed-upon values (but also potentially dangerous or undesirable) values.

**Positive Aspects:**
- Provides an interesting overview of existing work on training with synthetic data and a collection of potential future developments.
- Clearly presents potential harms and mitigation strategies.

**Negative Aspects:**
- As a short overview paper, the topics are not discussed in depth.
- In my view, the governance aspect remains somewhat ambiguous. The paper primarily presents possible good practices rather than politically actionable recommendations.

---

### Decision · Program_Chairs · 2025-03-04

Reject